# Sexual Assault Is the Biggest Risk Factor for Violence against Women in Taiwan—A Nationwide Population Cohort Study from 2000 to 2015

**DOI:** 10.3390/ijerph19063473

**Published:** 2022-03-15

**Authors:** Miao-Ju Chwo, Shi-Hao Huang, Yao-Ching Huang, Iau-Jin Lin, Chia-Peng Yu, Chi-Hsiang Chung, Wu-Chien Chien, Chien-An Sun, Gwo-Jang Wu

**Affiliations:** 1Department of Nursing, College of Medicine, Fu-Jen Catholic University, New Taipei City 242062, Taiwan; 071471@mail.fju.edu.tw; 2Department of Chemical Engineering and Biotechnology, National Taipei University of Technology (Taipei Tech), Taipei 10608, Taiwan; hklu2361@gmail.com (S.-H.H.); ph870059@gmail.com (Y.-C.H.); 3Department of Medical Research, Tri-Service General Hospital, Taipei 11490, Taiwan; iaujinlin@gmail.com (I.-J.L.); g694810042@gmil.com (C.-H.C.); 4School of Public Health, National Defense Medical Center, Taipei 11490, Taiwan; yu6641@gmail.com; 5Graduate Institute of Life Sciences, National Defense Medical Center, Taipei 11490, Taiwan; 6Taiwanese Injury Prevention and Safety Promotion Association (TIPSPA), Taipei 11490, Taiwan; 7Department of Public Health, College of Medicine, Fu-Jen Catholic University, New Taipei City 242062, Taiwan; 040866@mail.fju.edu.tw; 8Big Data Center, College of Medicine, Fu-Jen Catholic University, New Taipei City 242062, Taiwan; 9Department of Obstetrics and Gynecology, Tri-Service General Hospital, Taipei 11490, Taiwan; 10Graduate Institute of Medical Sciences, National Defense Medical Center, Taipei 11490, Taiwan

**Keywords:** sexual assault, violence, female, cohort study

## Abstract

Objective: To understand the main types of risk of violence against women in Taiwan. Materials and methods: This study used the outpatient, emergency, and hospitalization data of 2 million people in the National Health Insurance sample from 2000 to 2015. The International Classification of Diseases, Ninth Revision diagnostic N-codes 995.5 (child abuse) and 995.8 (adult abuse) or E-codes E960–E969 (homicide and intentional injury by others) were defined as the case study for this study, and the risks of first violent injury for boys and girls (0–17 years old), adults (18–64 years old), and elders (over 65 years old) were analyzed. Logistic regression analysis was used for risk comparison. A *p* value of <0.05 was considered significant. Results: The proportion of women (12–17.9 years old) who were sexually assaulted was 2.71 times that of women under the age of 12, and the risk of sexual assault for girls and adult women was 100 times that of men. Girls who were insured as labor insurance, farmers, members of water conservancy and fishery associations, low-income households, and community insured population (public insurance as the reference group) were significantly more likely to seek medical treatment from sexual assault than adult women. Among them, the risk was greatest for girls from low-income households (odds ratio = 10.74). Conclusion: Women are at higher risk of sexual assault than men regardless of whether they are children or adults, and the highest risk is for women in senior high schools, especially for girls from low-income households. Therefore, the protection of women’s personal autonomy is the direction that the government and people from all walks of life need to continue to strive for. Especially for high school students from low-income households, protection must be strengthened through education, social work, and police administration.

## 1. Introduction

The World Health Organization (WHO) defines partner and non-partner sexual violence separately, with partner sexual violence being defined as the self-reported forced engagement in sexual activity by a current or ex-partner from age 15 despite their unwillingness due to fear that their partner might act unfavorably during sexual intercourse or being forced to do something that is humiliating or degrading; non-partner sexual violence is defined as being 15 years of age or older when someone other than a person’s husband/partner is forced to perform any unwanted sexual act [1]. The revelation of sexual violence often creates shame and stigmatization of the victim; the perpetrator shames and blames the victim to reduce their responsibility, and a climate of stigma in sociocultural perceptions develops; in this case, most victims opt not to report their experiences or may not describe what happened to them as sexual violence [2]. WHO defines sexual abuse during childhood and adolescence (child sexual abuse (CSA)) as, “the involvement of a child in sexual activity that he or she does not fully comprehend, is unable to give informed consent to, or for which the child is not developmentally prepared and cannot give consent, or that violates the laws or social taboos of society; CSA is evidenced by this activity between a child and an adult or another child who by age or development is in a relationship of responsibility, trust, or power, the activity being intended to gratify or satisfy the needs of the other person” [3].

An important issue of sexual violence is the relationship between the victim and the perpetrator, and recent research has focused on the sexual violence between intimate partners, whether committed by a partner or a non-partner to the victim. Often traumatic, the pattern, extent, and effects of violence may vary by perpetrator [1,2,3]. The occurrence of spousal violence depends on determinants at the individual and environmental levels, with unemployment, poverty, and literacy having a significant impact on spousal violence against women [4]. Transgender and non-binary youths are exposed to significantly more violence compared to women and men. Experiences of sexual risk taking and ill health demonstrated strong associations with exposure to multiple violence [5].

Although most previous research has focused on the impact of domestic violence on women, a few studies have focused on the characteristics of adolescent girls and adult women who experienced sexual violence [6]. In Taiwan, no in-depth study has been conducted on this issue. Therefore, we hypothesized that sexual assault is the biggest risk factor for violence against women in Taiwan. This study intended to understand for the first time the main types of risk of violence against women in Taiwan through the National Health Insurance Research Database (NHIRD).

## 2. Materials and Methods

### 2.1. Data Source

Taiwan’s National Health Insurance launched a single-payment system on 1 March 1995. As of 2017, 99.9% of Taiwan’s population had participated in the program. This study was a 16-year observational research and used the NHIRD to provide a representative NHIRD 2000 coverage sample of 2 million people for the parent cohort (Longitudinal Health Insurance Research Database, LHID2000) as the research data source, tracking data on new cases for 16 years from 1 January 2000 to 31 December 2015. The files used were “Outpatient Prescription and Treatment Details File”, “Inpatient Medical Expense List Details File”, and “Insurance Information File”. Violent abuse research cases included 11,077 people. The National Institutes of Health encrypted all personal information before releasing the LHID2000 to protect the privacy of patients. In LHID2000, the disease diagnosis code was based on the “International Classification of Diseases, Ninth Revision, Clinical Modification” (ICD-9-CM) N-code standard. Cases that occurred in 2000 were excluded. Figure 1 shows the research-design flow chart of this study.

All procedures involving human participants performed in the research complied with the ethical standards of the institution and/or the National Research Council and the 1964 Declaration of Helsinki and its subsequent amendments or similar ethical standards. All methods were carried out following relevant guidelines and regulations. The Ethical Review Board of the General Hospital of the National Defense Medical Center (C202105014) approved this study.

### 2.2. Participants 

Defined children and adolescents who have suffered from violence (victims of violence) refer to minors under the age of 18 and who have joined the National Health Insurance for medical treatment. The scope, according to the International Classification of Diseases, Ninth Revision, Clinical Modification (ICD-9 CM) N-code: 995.5 and the definition of the external classification codes (E-code): E960–E969, of violently abused adult includes those who are 18–64 years old, whereas violently abused elderly refers to those 65 years of age and above, according to the ICD-9 N-code: 995.8 and E-codes E960–E969, as the case group (victims of violence). The control group consisted of people who did not suffer from violence (victims of violence). People in the case and control groups were matched in terms of index date, gender, and age at the ratio of 1:4.

The insured identity information came from the “unit attribute” variable of the underwriting file. The grouping method considered the original code, the data science center results’ carry-out requirements, and actual data distribution. The cases were divided into seven groups, namely, Group 1: “public insurance”; Group 2: “labor insurance”; Group 3: “Farmers”; Group 4: “Members of Water Conservancy and Fisheries Association”; Group 5: “Low-income households”; Group 6: “Community Insured Population”; Group 7: “Other + Missing Values”; Group 7: “Others” (including religious people, other social welfare institutions, veterans, and others).

The cause of injury in this study was identified in E-codes 960–969 (see Appendix A for details). The groups were combined based on the number of people that complied with the regulations of the Data Science Center of the Ministry of Health and Welfare. The main groups after grouping were, “Grapple, fighting, Sexual Assault (E960)”, “Injury by Cutting Tools (E966)”, “Children and Adults Persecuted and Abused (E967)”, and “Injured by Blunt Objects or Dropped Objects (E968.2)”. “Grapple, Fighting, Sexual Assault (E960)” was subdivided into “Unarmed combat or fighting—(E960.0)” and “Sexual Assault (E960.1)”. “Persecuted and abused children and adults (E967)” was subdivided into “Persecuted by father, stepfather or boyfriend (E967.0)” and “Persecuted by spouse or partner (E967.3)”, and the rest were classified as “Persecuted by others (E967.1), E967.2, E967.4–E967.9)”. The rest of the injuring methods were classified as “injured by other methods (E961–E965, E968.0, E968.1, E968.3–E968.7)”.

### 2.3. Statistical Analysis 

This study used the SAS 9.4 statistical software for Windows (SAS Institute, Cary, NC, USA) provided by the Academia Sinica Branch of the Data Welfare Center of the Ministry of Health and Welfare for the analysis. The descriptive statistics were expressed in the form of percentages, averages, and standard deviations, and the chi-square test was used to compare the differences among the three groups (children, adults, and elderly). Differences in the cause of injury and the proportion of women who suffered from sexual assault among different age groups were determined. In addition, logistic regression was used to analyze the risk of sexual assault for women in different age groups or with various occupations (including dependent occupations). According to the central limit theorem, (a) if the sample data are approximately normal, then the sampling distribution will also be normal; (b) in large samples (>30 or 40), the sampling distribution tends to be normal regardless of the shape of the data; (c) the means of the random samples from any distribution will themselves have a normal distribution [7]. A *p* value < 0.05 was considered to be statistically significant.

## 3. Results

During the 15-year period, 1592 children, 8726 adults, and 759 seniors were injured by violence and sought medical treatment. Among them, 301 children, 217 adults, and 0 seniors were sexually assaulted. Sexual assaults accounted for all the injuries. The proportions in each generation were 18.9%, 2.5%, and 0%, respectively, and the proportion of children suffering from sexual abuse was significantly higher than that of adults (Table 1). Although very few men were sexually assaulted, six cases occurred in childhood and five in adulthood.

Among female victims of violence, the proportions of injuries caused by sexual assault were 38.9%, 7.4%, and 0% in each generation; the proportion of those injured by unarmed combat or fighting rose to 24.5%, which was significantly higher than that of adult women, whereas 18.4% and 5.5% were observed for older women and girls, respectively (*p* < 0.0001) (Table 2).

The highest rate of sexual assault was observed among women 12–17 years old (54.8%), which is 2.71 times that of women under 12 years old. In addition, women aged 24–44 and 45–64 years old are more likely to be sexually assaulted than girls under 12 years old, who are less vulnerable to sexual assault (Table 3).

Girls and adult women were 100 times more likely to be sexually assaulted than men (*p* < 0.001). Senior age students (12–17 years old) were 2.5 times more likely to be sexually assaulted than junior age students (6–11 years old) (*p* = 0.003). For children and adolescents, and adults who were sexually assaulted, the risks of aggression were 11.4 (*p* < 0.001) and 2.51 times (*p* < 0.001) higher than that of the elderly, respectively (Table 4).

Girls who were insured as labor insurance, farmers, members of water conservancy and fishery associations, low-income households, and community insured population (public insurance as the reference group) were significantly more likely to seek medical treatment from sexual assault than adult women. Among them, the risk was the greatest for girls from low-income households (OR = 10.74) (Table 5).

## 4. Discussion

### 4.1. Importance of This Study

The results of this study revealed that the children and adolescents suffering from violence and seeking medical treatment accounted for the largest proportion, and the proportions of children and adolescents suffering from sexual abuse were significantly higher than that of adults. The proportions of children and adolescents suffering from sexual abuse were the majority. Women aged 12–17 years old were 2.71 times more likely to be sexually assaulted than women under 12. High school students (aged 12–17 years old) were 2.5 times more likely to be sexually assaulted than primary school students (aged 6–11 years old). Young people (18–23 years old) and adults (24–44 years old) were 11.4 and 2.51 times more likely to be sexually assaulted than middle-aged people (45–64 years old), respectively. The risk of sexual assault for girls in low-income households is greater than that of adult women (OR = 10.74). Therefore, sexual assault is the biggest risk factor for violence against women in Taiwan.

The most common forms of violence against women are domestic abuse and sexual violence [8]. Nearly 3 in 4 children or 300 million children aged 2–4 years regularly suffer from physical punishment and/or psychological violence at the hands of parents and caregivers. Exactly 1 in 5 women and 1 in 13 men reported having been sexually abused as a child at the age of 0–17 years. A total of 120 million girls and young women under 20 years of age have suffered from some form of forced sexual contact [9]. A study of grade 10 students in Iceland showed that 15% of them experienced some form of abuse, and two-thirds experienced abuse more than once [10]. A Swiss study noted that 40% of girls and 17% of boys reported CSA [11]. In a Swedish study, 65% of girls and 23% of boys reported CSA [12], which is consistent with our study.

Numerous studies have demonstrated the impact of poverty or low socioeconomic status (SES) on adolescent development and well-being [13,14,15]. A recent report from the Health Behavior in School-Aged Children study showed that disparities in household affluence continue to have a significant impact on adolescent health and well-being [16]. These findings suggest that adolescents from low-income households have poorer health, lower life satisfaction, higher levels of obesity and sedentary behaviors, weaker communication with parents, less social interaction through social media, and less social interaction from friends and family [17]. Many of these inequalities will have lasting lifelong effects. The findings suggest that these inequalities may be increasing, with widening disparities in several key areas of adolescent health [16,17].

In regard to sexual abuse in adolescents, a few studies have focused on the relationship between economic status (poverty or affluence) and CSA, and the results have been inconsistent [18]. The research has found poverty to be a risk factor for sexual abuse, Sedlak et al. reported that children from families with low SES were twice as likely to experience sexual abuse and three times as likely to be endangered than children from families with higher SES [19]. In their recent study, Lee et al. reported a high risk of severe and multiple types of abuse, including sexual abuse, for children experiencing poverty during childhood. This condition also affects the overall health in adult years, especially for women [20]. However, Oshima et al. found no significant difference in the CSA rates between more affluent and poor families, but a significant difference was reported between poor victims and wealthier victims of childhood sexual abuse for repeated reports of maltreatment to child protective services [21]. A few studies have looked at sexual abuse in low-income households and adolescence. Several research have shown that the least affluent adolescents reported a higher risk of sexual abuse [19,20], whereas one study reported no significant difference in CSA rates between non-poor and poor households [20,21]. Differences in these findings may stem from the differences in research methodology given that Oshima et al.’s data were derived from CSA reports from child protective services [19,20,21].

A low SES is an indicator of social disadvantage; for women, it may independently lead to the risk of sexual abuse. The double-harm hypothesis proposes that two or more concurrent sources of social disadvantage may interact to produce particularly negative outcomes. Therefore, the detrimental effects of SES may be more effective in adolescent girls than in boys [22]. The results of this study support this line of thinking.

From the perspective of violent criminology, countries attempt to prevent violence against women by formulating laws related to sexual assault [23]. However, the ineffectiveness of the law and the question of appropriateness still cannot effectively prevent women from suffering from violence; The Domestic Abuse Act of 2021 expanded the legal system’s role in dealing with domestic violence, made common assault an arrestable offense for the first time, and strengthened civil laws related to domestic violence to ensure that common-law partners of any gender and couples of any gender who have never been married or do not live together receive the same non-harassment and work order as married individuals [23].

Young people are the most frequent victims of sexual violence, with 12% to 25% of girls and 8% to 10% of boys under the age of 18 being thought to experience sexual violence [24]. In addition, CSA is associated with an increased risk of dating violence in all three forms (psychological, physical, and sexual) among boys and girls [25]. Sexual violence is more likely to occur among young people, women, people with disabilities, and those who have experienced poverty, childhood sexual abuse, and substance abuse [26,27]. Parental addiction, parental mental illness, and exposure to domestic violence, both individually and cumulatively, have been associated with CSA [28].

The shocking incident of two women being kidnapped and murdered in Taiwan at the end of 2020 prompted the passage of the “Stalking and Harassment Prevention Law” [29,30]. Violence against women and girls, irrespective of their social status and cultural level, remains prevalent throughout the world [30]. Previous investigations in Taiwan have noted that sexual assault victims between the ages of 12 and under 18 were the most common age group in 2006–2015 [29] but did not specifically identify low-income households’ girls as the most at-risk group [30]. Our study compared the risk of sexual assault between girls and adult women and pointed out that the risk of sexual assault for girls from low-income households in Taiwan is 10.74 times that of adult women.

In Taiwan, according to the latest “Statistical Survey on Intimate Relationship Violence of Taiwanese Women” released by the Ministry of Health and Welfare, 20% of women have been subjected to violence by an intimate partner, of which mental violence is the most common, whereas sexual violence has doubled compared with previous surveys [30]. A slight increase has also been observed in harassment, which is a form of violence in intimate relationships and needs attention in the future [30]. In 2021, a woman was stalked and harassed in Taiwan [30]. When no legal basis and no way to seek help were found, an unfortunate incident finally occurred, which led to the passage of the third reading of the “Stalking and Harassment Prevention Act”, making Taiwan a legal basis for the protection of women’s rights and interests [30].

### 4.2. Cause of High Risk of Sexual Assault among High School Girls

This issue needs to be discussed from the criteria for determining sexual abuse. The condition for CSA must be that the child does not have a genuine consent [31]; however, the consensual behavior of boys and girls in the case of mutual consent still constitutes a constitutive element of sexual assault in terms of legal standards [31]. Therefore, when medical personnel in Taiwan are faced with sexual assault cases under the age of 18, they are required to log in the sexual assault code and report according to law [31].

Previous studies indicated that adolescents who suffered from sexual assault were mostly younger than 14 years old, whereas this research showed that the high-risk group for sexual assault included high school girls aged 13–17 years old, which is consistent with the female sexual maturity age [32]. The Swedish survey revealed that sexual violence accounted for 16.3% before the age of 18, and 10.2% of women experienced/attempted sexual assault in adulthood [33]. Perpetrators consisting of uncles and stepfathers were more common among adolescents and partners or ex-intimate partners of adult women; in most cases, sexual assaults occurred in public places, although sex crimes at the perpetrator’s residence were more frequent among adolescents [32]. The 2008–2020 Sexual Assault Notification Case Investigation in Taiwan provides the age distribution of sexual assault victims and perpetrators. Over the years, most of the victims were 12 to under 18 years old, and most perpetrators were 12 to under 18 years old and 18 to under 24 years old [34].

Feminist scholars reject biological and essentialist explanations, arguing that gender inequality is the driving force behind sexual violence against women [35]. Sanday, who first proposed the theoretical framework of sexual violence, believed that sexual assault was used as a means to control and dominate women to maintain the hierarchical status of men [36]. However, such a theoretical framework cannot fully explain the difference in the risk of sexually assaulted girls and adult women. In the work of, a more reasonable explanation can be obtained from the following three factors: (1) low-income girls face more capable and criminally motivated offenders than adult women; (2) low-income girls and adult women are more suitable targets for sexual violence crimes; (3) girls from low-income households are more likely to face the absence of suppressors who can stop the crime [37]. When the above three conditions all develop in an unfavorable direction, the risk of sexual assault for girls in low-income households is 10.74 times that of adult women, and the risk of sexual assault for girls in other insurance statuses is also higher than that of adult women.

This study has several limitations. First, Taiwan’s National Health Insurance database employs the practice of delaying the release of data for two years. Moreover, from 2016 to 2018, the data will face the problem of changing the data code from ICD9 to ICD10, which may cause a deviation in the code conversion. Second, Taiwan’s National Health Insurance database lacks information on personal factors, such as marriage, education level, and living habits. The problem of child marriage is not evident in Taiwan, and in Taiwanese women’s secondary education (12–17 years old) from 2000 to 2015, the rate was 93.49–96.28%. Thus, the lack of the above variables had little impact on this study. Third, the occupational classification of the health insurance database is not in accordance with the classification required for the research, and a more detailed classification cannot be obtained. However, the identification of low-income status is recognized by the relevant Taiwan authorities and has credibility. However, researchers cannot avoid the low-income status. Finally, after years of promotion in Taiwan, the prevention and treatment of sexual assault in Taiwan now bears a standard medical procedure, and child sexual assault is now a public prosecution crime. Therefore, despite the possibility of bias, the researchers believe that the associated range is small.

## 5. Conclusions

Our results showed that regardless of whether women are children or adults, the risk of sexual assault is higher than that of men, and women in national high schools are at the highest risk, especially girls from low-income households. These results highlight the vulnerability of children, especially women, living in low-income households to CSA. They also underscore the urgency of financially supporting the children of these low-income households, given the severity of the impact of CSA on the future health and well-being of victims. Therefore, the protection of women’s personal autonomy is the direction that the government and people from all walks of life need to continue to strive for. Politicians and health professionals, welfare, and education play an important role in supporting low-income children and their families. For high school students from low-income households, their protection must be strengthened through education, social work, and police administration.

Future studies should compare the impact of pre-coronavirus disease (COVID-19) versus post-COVID-19 sexual violence issues, such as those in 2016–2020 versus 2020–2024, after the update year is released given that COVID-19 is expected to exacerbate this phenomenon.

## Figures and Tables

**Figure 1 ijerph-19-03473-f001:**
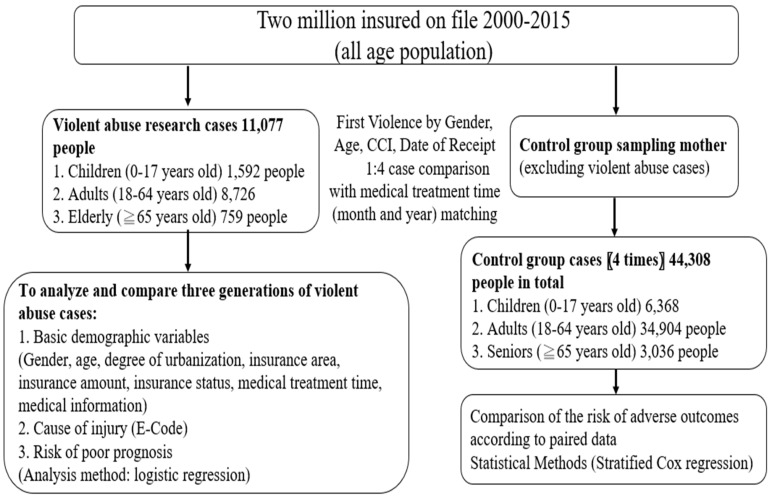
Research flowchart of poor prognosis in cases of violent abuse.

**Table 1 ijerph-19-03473-t001:** Causes of injury to victims of violence in different generations.

Cause of Injury (E-Code E960–E969) ‡	Children and Adolescents (*n* = 1592)		Adult (*n* = 8726)		Elderly (*n* = 759)		
	*n*	%	*n*	%	*n*	%	*p*
Grapple, fighting, sexual assault (E960)	659	41.4	3265	37.4	183	24.1	<0.0001 **
Unarmed combat or fighting (E960.0)	315	19.8	2877	33.0	173	22.8	<0.0001 **
Sexual Assault (E960.1)	301	18.9	217	2.5	0	0.0	<0.0001 **
Damage from cutting tools (E966)	94	5.9	803	9.2	57	7.5	<0.0001 **
Persecuted and abused children and adults (E967)	73	4.6	550	6.3	43	5.7	0.0273 *
Persecuted by father, stepfather, or boyfriend (E967.0)	16	1.0	22	0.3	0	0.0	<0.0001 **
Persecuted by spouse or partner (E967.3)	0	0.0	232	2.7	8	1.1	<0.0001 **
Persecuted by others (E967.1, E967.2, E967.4–E967.9)	56	3.5	293	3.4	35	4.6	0.1927
Wounded by blunt object or dropped object (E968.2)	95	6.0	705	8.1	46	6.1	0.0034 **
Harmed by other means (E961–E965, E968.0, E968.1, E968.3–E968.7)	475	29.8	3447	39.5	382	50.3	<0.0001 **

Tested by chi-square test, *: *p* < 0.05, **: *p* < 0.01. ‡ Check.

**Table 2 ijerph-19-03473-t002:** Causes of Injury of Victims of Violence in Different Generations (Women).

Cause of Injury (E-Code E960–E969) ‡	Children and Adolescents (*n* = 759)		Adult (*n* = 2873)		The Elderly (*n* = 282)		
	*n*	%	*n*	%	*n*	%	*p*
Grapple, fighting, and sexual assault (E960)	357	47.0	981	34.1	58	20.6	<0.0001 **
Unarmed combat or fighting (E960.0)	42	5.5	705	24.5	52	18.4	<0.0001 **
Sexual Assault (E960.1)	295	38.9	212	7.4	0	0.0	<0.0001 **
Damage from cutting tools (E966)	17	2.2	147	5.1	17	6.0	0.0018 **
Persecuted and abused children and adults (E967)	31	4.1	357	12.4	18	6.4	<0.0001 **
Persecuted by father, stepfather, or boyfriend (E967.0)	6	0.8	16	0.6	0	0.0	0.3160
Persecuted by spouse or partner (E967.3)	0	0.0	206	7.2	5	1.8	<0.0001 **
Persecuted by others (E967.1, E967.2, E967.4–E967.9)	24	3.2	132	4.6	13	4.6	0.2182
Wounded by blunt object or dropped object (E968.2)	12	1.6	113	3.9	17	6.0	0.0007 **
Harmed by other means (E961–E965, E968.0, E968.1, E968.3–E968.7)	100	13.2	885	30.8	141	50.0	<0.0001 **

Tested by chi-square test, **: *p* < 0.01. ‡ Check.

**Table 3 ijerph-19-03473-t003:** Basic demographic data of child and adult female sexual assault victims (by cause).

Demographic Variables		Sexually Assaulted		Other Reasons				
		*n*	%	*n*	%	*p* ^a^	OR (95% CI)	*p* ^b^
Age (years)	<12	17	3.4	66	2.1	<0.0001 **	1.000	
	12–17	278	54.8	398	12.7		2.71 (1.56–4.72)	0.0004 **
	18–23	97	19.1	331	10.6		1.14 (0.64–2.03)	0.6623
	24–44	93	18.3	1480	47.4		0.24 (0.14–0.43)	<0.0001 **
	45–64	22	4.3	850	27.2		0.10 (0.05–0.20)	<0.0001 **

^a^: Tested by chi-square test, ^b^: Analyzed by logistic regression **: *p* < 0.01.

**Table 4 ijerph-19-03473-t004:** Relative risk of sexual assault victims (by age).

Age Group	OR (95% CI)	*p*
Senior age vs. Junior age	2.50 (1.37–4.55)	0.0030 **
Children and Adolescents vs. The Elderly	11.4 (7.07–18.4)	<0.0001 **
Adult vs. The Elderly	2.51 (1.57–4.01)	0.0001 **

Analyzed by logistic regression, **: *p* < 0.01, gender adjusted.

**Table 5 ijerph-19-03473-t005:** Relative risk for victims of sexual assault (by occupation).

Occupation of the Person or the Health Care Dependent	Girls vs. Adult Women	
	OR (95% CI)	*p*
Public insurance	1.00	
Labor insurance	4.87 (1.31–18.13)	0.0184 *
Farmer	4.44 (1.04–19.01)	0.0443 *
Member of Water Conservancy and Fisheries Association	5.93 (1.29–27.28)	0.0224 *
Low-income households	10.74 (2.42–47.72)	0.0018 **
Community insured population	3.91 (1.04–14.79)	0.0443 *

Analyzed by Logistic regression, *: *p* < 0.05, **: *p* < 0.01.

## Data Availability

Data are available from the NHIRD published by the Taiwan NHI administration. Because of legal restrictions imposed by the government of Taiwan concerning the “Personal Information Protection Act”, data cannot be made publicly available. Requests for data can be sent as a formal proposal to the NHIRD (http://www.mohw.gov.tw/cht/DOS/DM1.aspx?f_list_no=812 (accessed on 13 October 2021)).

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
