# Peer review of "Sexual Assault Is the Biggest Risk Factor for Violence against Women in Taiwan—A Nationwide Population Cohort Study from 2000 to 2015"

_ijerph, 2022, doi:10.3390/ijerph19063473_

Round 1
Reviewer 1 Report
The topic is really interesting and the data collected is impressive as size sample, but I think if possible you need to update it. 2015 is already 7 years ago and 5 years prior COVID, so even if your recommendations are highly necessary, they must be supported by newer data (because you are talking about COVID and its effects in the discussions). Another example that needs an update is that maybe the laws have suffered some changes in the last years, you mention some documents from 2004. Also, the whole discussions should focus more on the results of your analysis, because in my opinion, in this form the ideas are very general and you really have some very interesting results to comment there. On the conclusions chapter, the ideas should be more developed and the tendencies you have revealed could be linked to some practical recommendations.
Overall, this study has a good sample of subjects and interesting results that could be exploited more, in order to obtain a the best of it.
Author Response
See uploading file.

Reviewer 2 Report
Thank you for the manuscript, that addresses a global problem. The following needs to be addressed:
- The abstract is too long and should only provide the most evident results.
- The objective in the abstract and main text should be aligned: Is the study objective "to understand the main types of risk of violence against women in Taiwan" or "to explore the relevant epidemiological characteristics of women who are victims of violence and identify the high-risk groups of women who are victims of violence"?
- Some paragraphs in the methods section is written in the present tense or future tense while it has already happened eg the ethical aspects.
- Some of the groups are not well phrased eg, what is meant by "fighting, fighting, Sexual Assault"?
- Under results: "We screened 24,363 obese patients (cases) and 97,452 non-obese patients (control group). " - not clear what obseity has to do with the study topic?
- The age categories under which results are discussed are not similar and consistently followed, for example, "Children and Adolescents, Adult, The Elderly" and then "Senior age vs Junior age, Youth vs Middle age, Adult vs Middle age". This makes it very difficult to follow and interpret the results in terms of age groups.
- The use of the sentence about COVID in the discussion is not clear as the study was done before COVID: "Among the measures implemented by different countries to contain the number of infections and deaths from the COVID-19 pan-demic recommended by experts and epidemiologists, in order to stem the spread of COVID-19, forcing people to stay at home for longer and interact with their families, from isolating"
- The discussion section is very long and provides an overview of violence aganist women, statistics and prevalence, consequences and interventions, as well as legislation. The focus should rather be on the study results, comparing those with similar results from othe countries. Some of the sentences on prevalence can rather be moved to the introduction to show the extent of the problem. The second section of the discussion is more applicable as it covers risk factors.
- The conclusion should also provide research recommendations emerging from the study results.
- The manuscript is not well written and some sentences are not clear with punctuation and grammar mistakes. Therefore some sections cannot be interpreted and commented upon as it is impossible to detect what is meant and follow the logical flow of the discussion.
Author Response
See uploading file

Reviewer 3 Report
The present manuscript deals with a current and relevant theme in the world scenario; the work is well written and can be published in this journal. However, I suggest to the authors some notes that can improve the quality of the manuscript.
At the end of the introduction, I recommend inserting the research hypotheses more clearly because they are not in the text. Don't forget to bring them again to the "discussion" topic.
Regarding the method, although the authors have inserted the inclusion criteria for the participants, you also need to insert the exclusion criteria. It was unclear to me how the participants were recruited, which needs to be presented in the text.
The authors could also describe the data analysis in more detail, as some statistical components are important for the readers to understand these analyses' steps.
Finally, I recommend that the authors insert future directions for further studies in the text's conclusion.
Author Response
See uploading file.

Round 2
Reviewer 2 Report
The authors implemented the reviewer recommendations, thank you for the comprehensive report
Reviewer 3 Report
The authors made all the requested changes.